# Chemical Topping with Mepiquat Chloride at Flowering Does Not Compromise the Maturity or Yield of Cotton

Haikun Qi [1,2,†], Chenyu Xiao [1,2,†], Wenchao Zhao [3], Dongyong Xu [4], Anthony Egrinya Eneji [5], Zhengying Lu [6], Rongrong Shao [7], Guifeng Wang [8], Mingwei Du [1,2], Xiaoli Tian [1,2,*] and Zhaohu Li [1,2,*]

1   Engineering Research Center of Plant Growth Regulator, Ministry of Education, China Agricultural University, Beijing 100193, China
2   College of Agronomy and Biotechnology, China Agricultural University, Beijing 100193, China
3   Dezhou Academy of Agricultural Sciences, Dezhou 253015, China
4   Hebei Cottonseed Engineering Technology Research Center, Hejian 062450, China
5   Department of Soil Science, Faculty of Agriculture, Forestry and Wildlife Resources Management, University of Calabar, Calabar 1115, Nigeria
6   Early Maturing Cotton Research Institute Handan Academy of Agricultural Sciences, Handan 056000, China
7   Binzhou Academy of Agricultural Sciences, Binzhou 266000, China
8   Shandong Province Agro-Tech Extension and Service Center, Jinan 250100, China
*   Correspondence: tianxl@cau.edu.cn (X.T.); lizhaohu@cau.edu.cn (Z.L.); Tel.: +86-10-62734550 (X.T.); +86-10-62734550 (Z.L.)
†   These authors contributed equally to this work.

**Abstract:** The balance between vegetative and reproductive growth is the central objective in the cotton production system, which is associated with cotton maturity and yield. In China, manual topping (MT) has been performed many years prior to or during the flowering period to inhibit vegetative growth and enhance reproductive growth. MT is gradually being replaced by chemical topping (CT) with mepiquat chloride (MC, 180 g ha$^{-1}$, 98% soluble powder) due to labor shortages and increasing labor cost. To determine whether CT influences cotton maturity and yield relative to MT, we carried out field experiments at four locations in the Yellow River Valley of China during 2018–2020. The results showed that CT did not alter the boll age, and although it produced taller and slender spatial boll distribution under several environments, it had little effect on the accumulation of boll fraction (the number of bolls in a given period divided by total boll number) over time at the end of blooming across locations. As a result, there were no significant differences between MT and CT in boll opening percentage in the late season. CT did not influence yield or yield components, except under severe drought. Therefore, CT with MC (180 g/ha, 98% soluble powder) during the flowering period will not compromise the maturity or yield of cotton in the Yellow River Valley of China. Similar outcomes would be achieved in other areas with similar ecological conditions and social conditions that require an alternative to extensive manual labor.

**Keywords:** mepiquat chloride; cotton maturity; Yellow River Valley of China; boll distribution; plant mapping





## 1. Introduction

Cotton has an indeterminate growth habit in which both vegetative and reproductive growth occur at the same time. Too small a vegetative structure on the plant results in reduced yield potential, and too much vegetative development compromises fruit set and yield. Accordingly, the central objective in a cotton production system is to provide a well-balanced level of development between the vegetative structure and the crop's fruit, hence the vegetative/reproductive balance [1].

In addition, a well-balanced vegetative/reproductive growth will result in earliness. In general, when boll set is continuous over time, the plant stops producing new fruit due to demand on the assimilate supply by the developing fruit, leaving none for the initiation

of new fruiting sites [2]. However, in good growing conditions (moderate temperatures, good soil moisture, and adequate fertility), cotton can maintain the production of new fruit for an extended time. As a result, not all cotton fruit is mature at harvest time, thus leading to late maturity. Harvesting a mature crop is essential for maximizing yield, crop quality, and economic return [3].

Many production strategies and management decisions are usually made to balance vegetative/reproductive growth. Mepiquat chloride (MC) is an anti-gibberellin growth retardant that reduces plant cell enlargement to control rank growth by reducing stem elongation at newly formed internodes, and thus helps balance vegetative and reproductive growth [4–6]. In addition, because of the shorter growing seasons in China (about 2500 DD60s vs about 2600 DD60s in the cotton belt of the US) and thus less concern about regrowth caused by manual topping (MT), the manual removal of the top of the main stem (i.e., MT) during the bloom period is the traditional method of transitioning cotton from vegetative to reproductive growth [7]. However, due to recent shortages and the high cost of labor, MT in China is being replaced by an extra MC application in addition to the conventional multiple application of MC simultaneously with MT, which is known as chemical topping (CT) [5,8,9].

Many studies have shown that MC applications can hasten cotton maturity by enhancing retention of early buds and bolls, as well as reducing boll age [10–14]. However, whether CT with an extra MC application (in addition to regular multiple MC applications) during flowering would affect cotton maturity compared with MT (also experiencing regular multiple MC application) remains unclear. Theoretically, a common effect of MT and CT is the inhibition of vegetative growth of cotton plants. However, MT can eliminate the apical dominance immediately and stop the formation of new nodes, whereas CT only weakens the apical dominance gradually, possibly allowing plants to produce more fruiting branches, and the vegetative growth may be more vigorous than that under MT in some periods. Thus, we hypothesized that CT may delay maturity relative to MT.

Plant mapping of fruit locations by main-stem node and sympodial fruiting position within the plant canopy is an important approach to estimating crop maturity [14,15]. Usually, bolls in the lower fruiting branches and inner fruiting nodes have more heat resource to mature [16]. Therefore, the more bolls in these parts, the more likely the plant matures early.

Few studies have investigated the effects of CT on boll distribution and cotton maturity. Zhang et al. [14] showed that there were no significant differences between CT and MT in the N50 values (the node at which 50% of the total yield is derived) and pre-frost boll opening rate, but CT enhanced earliness by smaller N90 values. Li et al. [17] reported that CT enhanced maturity better than MT, characterized by higher percentage of boll opening in late September and higher ratio of yield at first harvest to total yield. According to Dai et al. [18], there were few differences in earliness (the percentage of the first two harvests to total harvest) between MT and CT regardless of plant density and ecological conditions. These contrasting reports suggest no certainty on the effects of CT on cotton maturity. In addition, the methods adopted in these previous studies to measure cotton maturity were not comprehensive. Most of them focused on end-of-season maturity and ignored the boll maturation period or accumulation of boll fraction over time. For studies involving use of defoliants [19,20], the ratio of first or first two harvests to total yield was likely influenced by the interaction between defoliants and CT.

In this study, a multiple-site field experiment was performed in the Yellow River region of China during 2018–2020. The boll age (boll maturation period), spatial boll distribution, boll accumulation by cohorts (each cohort containing individual bolls of nearly the same age) [21], and open boll percentage at defoliation were used to compare cotton maturity under MT and CT. The effects of CT on cotton yield were also investigated. The results would improve our understanding of the role and how to more efficiently use CT for the field management of cotton.

## 2. Materials and Methods

### 2.1. Location and Cultivar

The experiments were conducted at four locations (Figure 1) in the Yellow River region of China during the growing seasons of 2018–2020. These were Hejian (38°27′09.76″ N, 116°06′20.26″ E) and Handan (36°37′55.88″ N, 114°32′44.27″ E) in Hebei Province and Dezhou (37°26′27.06″ N, 116°22′4.27″ E) and Binzhou (37°46′45.81″ N, 117°37′42.37″ E) in Shandong Province. In 2018, the cultivar Xinkang 4 (developed by Hebei Cottonseed Engineering Technology Research Center, Hejian, Hebei Province) and Han 853 (developed by Handan Academy of Agricultural Sciences, Handan, Hebei Province) were used in Hejian and Handan, respectively, and SCRC 37 (developed by Cotton Research Center, Shandong Academy of Agricultural Sciences, Jinan, Shandong Province) was used in both Dezhou and Binzhou. Guoxin18-4 and Guoxin 26 (both developed by Hebei Cottonseed Engineering Technology Research Center, Hejian, Hebei Province) were used for all locations in 2019 and 2020. The major agronomic practices for cotton production in these locations are shown in Table 1.

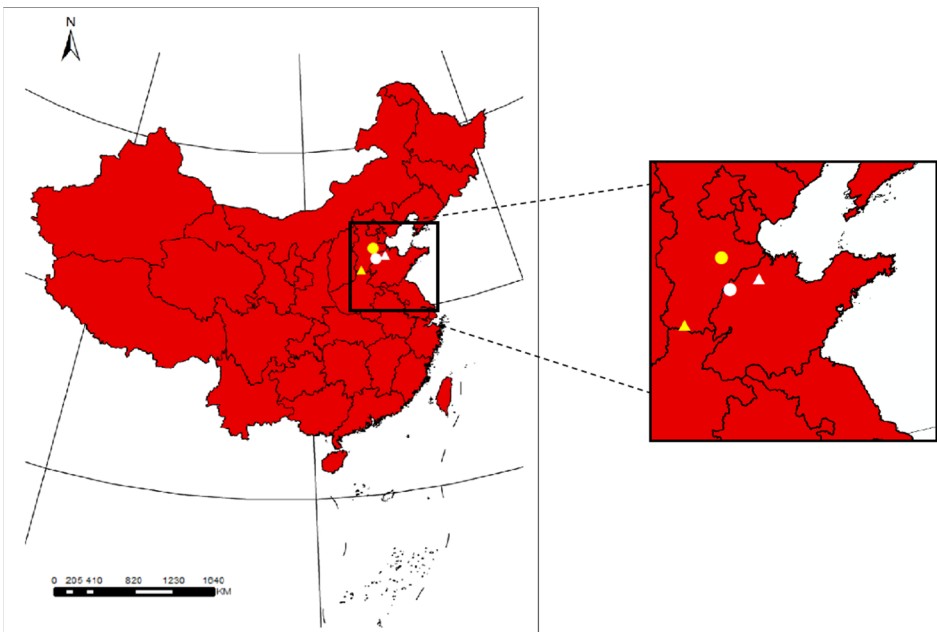

**Figure 1.** Location of the experiment sites. Yellow point and yellow triangle represent Hejian and Handan, Hebei Province; white point and white triangle represent Dezhou and Binzhou, Shandong Province.

The monthly average temperature and cumulative rainfall during the cotton growing season at each location are shown in Figure 2. Hejian was always cooler than other sites in September and October. There was an unusually rainy August in Binzhou and Dezhou, but an extremely dry August in Handan in 2018. In 2020, there was much less precipitation in July (<50 mm) except in Hejian (107.9 mm), but heavy rains in August (>210 mm) across locations. In addition, Hejian witnessed a cooler July and August besides September and October than the other three sites in 2020.



**Table 1.** Major agronomic practices for cotton production in the study locations.

| Year | Location | Variety | Row Spacing (cm) | Plant Density ($10^4$ Plant ha$^{-1}$) | Sowing Date | Manual Topping Date | Frequency of Regular MC * Application | Rate of Regular MC Application (g ha$^{-1}$) | Nitrogen Rate (kg ha$^{-1}$) | Date of Harvest Aids Applied |
|---|---|---|---|---|---|---|---|---|---|---|
| 2018 | Hejian, Hebei | Xinkang4 | 76 | 9.0 | April 27 | July 20 | 3 | 147.0 | 150 | — |
| | Handan, Hebei | Han853 | 66 | 5.3 | April 26 | July 17 | 1 | 120.0 | 0 | — |
| | Dezhou, Shandong | Lumianyan37 | 76 | 4.5 | April 28 | July 26 | 6 | 397.5 | 0 | — |
| | Binzhou, Shandong | Lumianyan37 | 76 | 6.0 | April 25 | July 25 | 0 | 0 | 150 | — |
| 2019 | Hejian, Hebei | Guoxin18-4 | 76 | 9.0 | May 1 | July 22 | 3 | 240.0 | 75 | September 23 |
| | Handan, Hebei | Guoxin18-4 | 76 | 9.3 | April 28 | July 16 | 1 | 30.0 | 150 | September 19 |
| | Dezhou, Shandong | Guoxin18-4 | 76 | 4.5 | April 29 | July 18 | 6 | 360.0 | 0 | September20 |
| | Binzhou, Shandong | Guoxin18-4 | 76 | 6.0 | May 9 | July 20 | 5 | 412.5 | 150 | September 22 |
| 2020 | Hejian, Hebei | Guoxin26 | 76 | 9.0 | April 24 | July 20 | 5 | 225.0 | 150 | September 16 |
| | Handan, Hebei | Guoxin26 | 76 | 9.3 | April 25 | July 14 | 5 | 300.0 | 150 | September 12 |
| | Dezhou, Shandong | Guoxin26 | 76 | 6.8 | April 27 | July 16 | 6 | 307.5 | 150 | September 25 |
| | Binzhou, Shandong | Guoxin26 | 76 | 7.5 | April 29 | July 17 | 5 | 300.0 | 150 | September 18 |

*: MC means mepiquat chloride, 98% soluble powder.

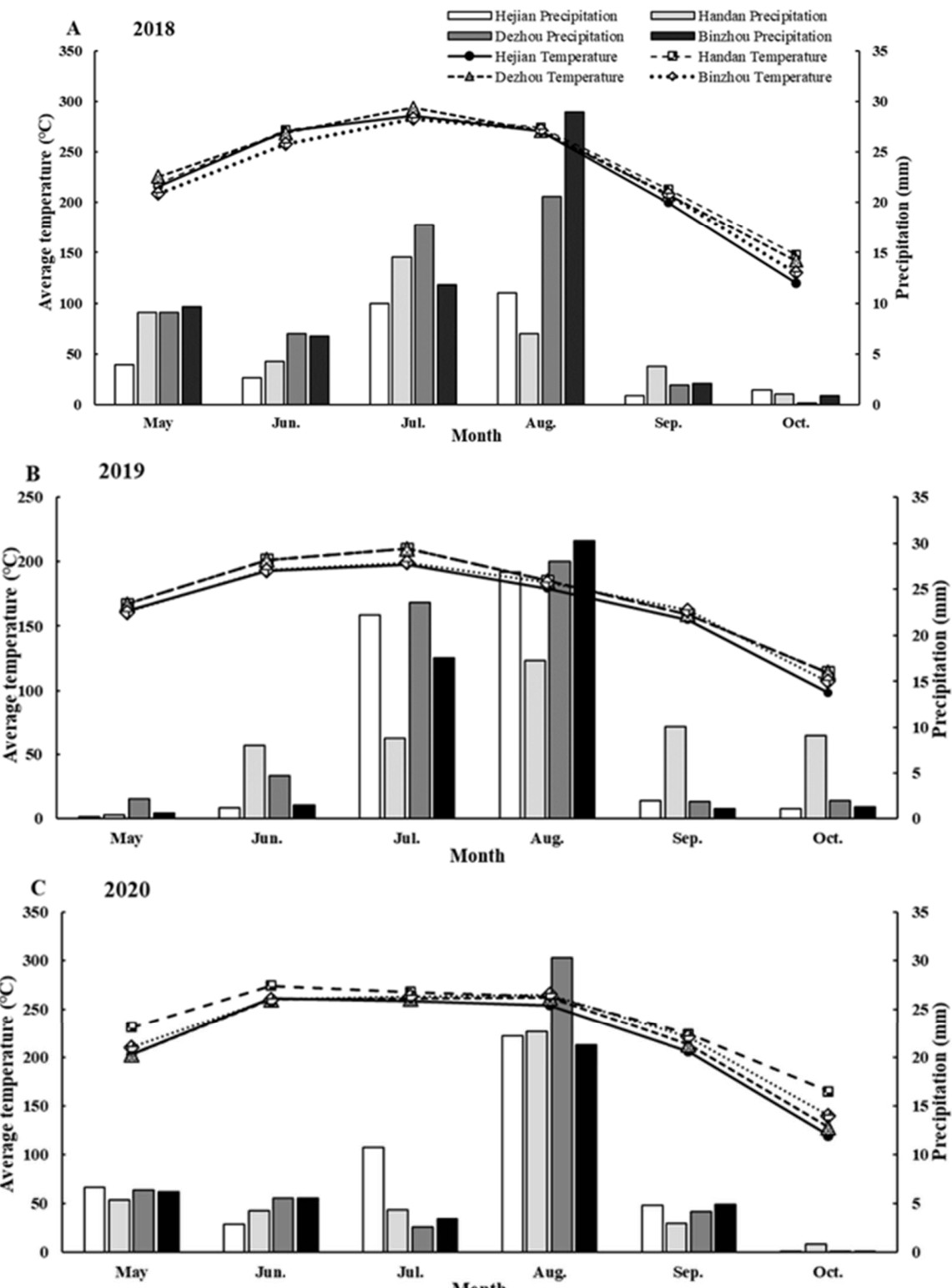

**Figure 2.** Monthly average temperature and cumulative precipitation during the cotton growing seasons in 2018 (**A**), 2019 (**B**), 2020 (**C**).

## 2.2. Experimental Design and Field Management

A randomized block design with three replications was used in this study. The experiment consisted of two treatments: manual topping (MT) and chemical topping (CT). The MT timing followed local tradition (Table 1), which is generally based on two rules: the

newly formed squares do not have enough heat for maturation or the number of fruiting branches has met expectation. CT was performed by applying 180 g/ha mepiquat chloride (MC, 98% soluble powder) at the same time as MT. The rates were determined by prior experiments and local practices [19].

Individual plots were 6–9 m long and comprised 6–8 rows, spaced 76 cm apart (except in Handan in 2018). The cotton was seeded from April 24 through May 9 (Table 1). The plant density and nitrogen (N) rates applied at each location are shown in Table 1. The overgrowth during season was controlled by multiple MC application (Table 1) according to local weather and environments. Plants were pruned of vegetative branches on June 20 and July 15 in Handan, and on June 14 and July 10 in Dezhou in 2018. In 2019, plants were pruned of vegetative branches on June 10 only in Dezhou. Pruning was not done in 2020. The management of weeds, diseases and insects followed local practices.

### 2.3. Measurement

### 2.3.1. Boll Age

The white flowers at the 1st and 2nd fruiting sites on middle fruiting branches were tagged in mid-July, 2019 and 2020. Around 50 days later, we counted the total and open tagged bolls to calculate boll opening percentage (BOP): higher BOP means lower boll age.

### 2.3.2. Plant Mapping

Five or ten uniform plants per plot were selected to perform plant mapping. Few bolls arise from vegetative branches, but usually the proportion of these bolls to total yield is low [22,23]. Therefore, plant mapping only covered the harvestable fruit by fruiting sites on reproductive branches.

### 2.3.3. Boll Accumulation over Cohorts

Flowers are formed on adjacent nodes up the plant every 2 to 3 days, and on adjacent sympodial positions every 4 to 6 days [24]. Based on both vertical and horizontal flowering intervals, the first-position fruit may occur simultaneously with second-position fruit that are two nodes lower and third-position fruit that are four nodes lower, and so on. These appear at the same time and constitute a cohort. In this study, we grouped bolls according to cohorts, and boll fraction was accumulated over it: higher accumulation over a given cohort indicates an earlier maturity.

### 2.3.4. Boll Opening Percentage (BOP) before Defoliation in September

The timing of defoliation using 50% thidiazuron·ethephon suspension concentrate (TDZ•ETH) is shown in Table 1. Before TDZ•ETH application, the number of total (NTB) and open bolls (NOB) were counted. Then, the percentage of boll opening was calculated: boll opening percentage (BOP, %) = NOB $\times$ 100%/NTB.

### 2.3.5. Yield and Yield Components

In late October or early November, fifteen plants with uniform growth were selected per plot to determine boll number and boll weight. In addition, 50 random bolls per plot were sampled to determine lint percentage. All plants in the central three rows of each plot were harvested to evaluate seed cotton yield.

### 2.4. Data Analysis

Microsoft Excel 2016 (Microsoft Corp, Albuquerque, NM, USA) was used for data organization. Because the varieties used at each location were different in 2018, a *t*-test was performed to compare the effects of MT and CT on boll age, boll opening percentage (BOP) in mid- or late September, cotton seed yield and yield components per location at $p \leq 0.05$. The general linear model procedure in R Core Team (2022) was used for analysis of variance (ANOVA) of the data for 2019 and 2020 when the same variety was used across

locations, and the least significant difference (LSD) was used to separate treatment means at the 5% level of probability.

Contour maps of boll distribution and box plots of boll accumulation by cohort were drawn using Sigma Plot 10.0 (Systat Software Inc., San Jose, CA, USA) and R Core Team (2022), respectively.

## 3. Results

### 3.1. Boll Age

Boll age indicates the period from white flower to boll opening. A short boll age enhances cotton earliness. In this study, a BOP around 50 days after blooming was used to indirectly estimate the boll age. A higher BOP suggests a younger boll. As shown in Figure 3, there were no significant differences in BOP between MT and CT for bolls set in mid-July (prior to or after peak bloom) regardless of year and location, indicating that CT did not influence the boll age.

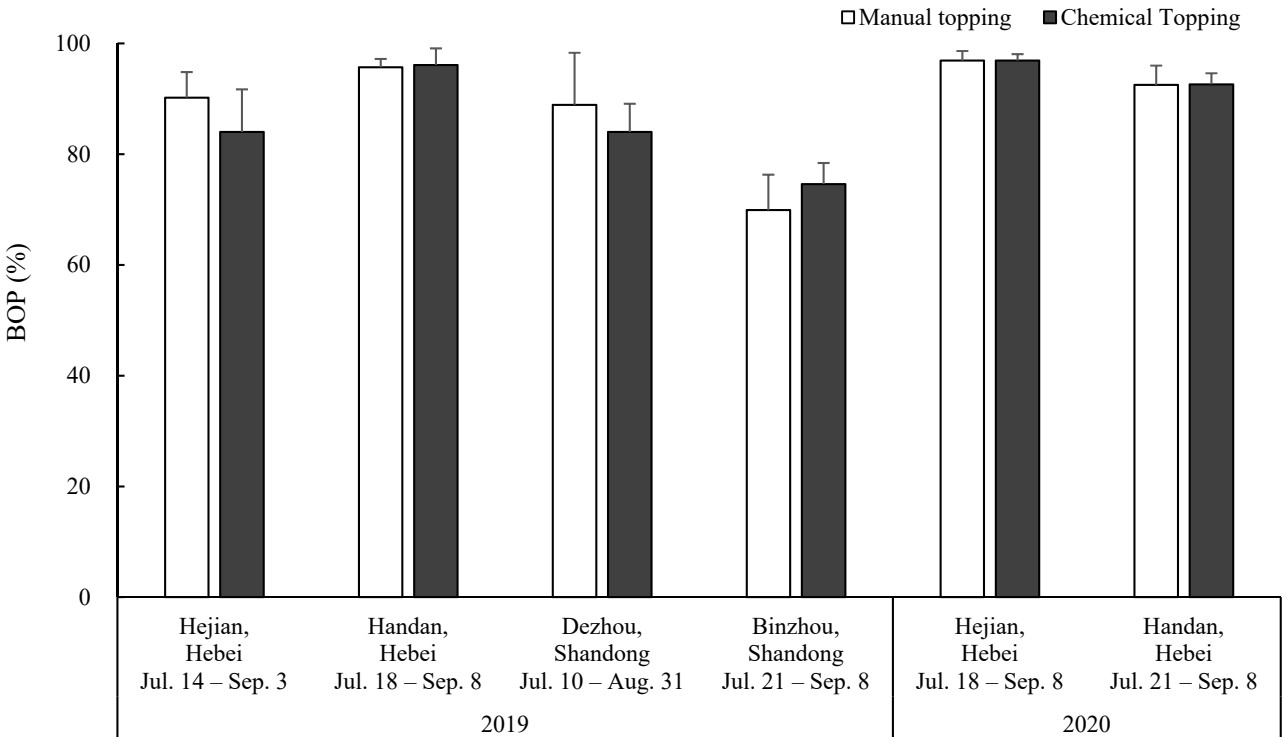

**Figure 3.** Effect of topping treatments on the opening percentage of tagged bolls (BOP). The young bolls were tagged on the first day of bloom in July, and BOP was determined after about 50 days in September; *t*-tests were used to compare the means for manual topping and chemical topping at the 0.05 probability level.

### 3.2. Spatial Distribution of Bolls

The boll distribution varied greatly with years and locations (Figure 4), suggesting the effects of environment, genotype and management on it. In most cases, CT had no or only slight effects on boll distribution (Figure 4). However, it produced taller but slenderer boll distribution than MT in Hejian in 2018 (Figure 4A) and in Hejian and Handan in 2019 (Figure 4I,J), indicating fewer bolls at outer parts of lower and middle fruiting branches, but more bolls at upper parts of plants.

### 3.3. Accumulated Fraction of Bolls by Bloom Cohort

Cotton fruit at different positions may bloom and thus probably mature at the same time. These fruit could be grouped into a cohort. For estimating maturity, the total boll

accumulation by cohort would be more useful than that by main-stem node, because the fruit derived from the same node can bloom and mature at different times.

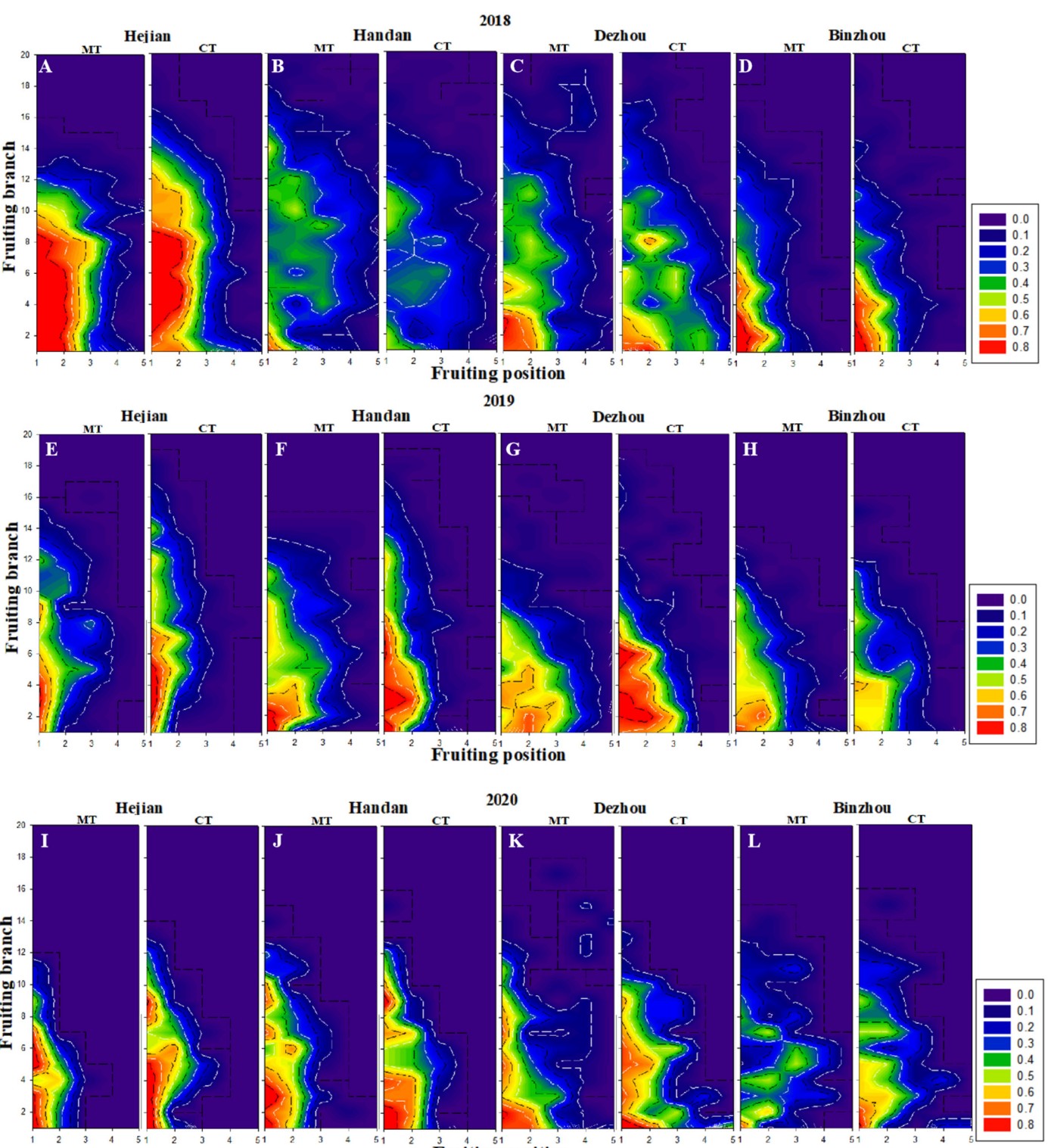

**Figure 4.** Effect of topping treatments during flowering on distribution of cotton bolls. MT and CT indicate manual topping and chemical topping, respectively. Hejian (**A**,**E**,**I**) and Handan (**B**,**F**,**J**) are in Hebei Province; Dezhou (**C**,**G**,**K**) and Binzhou (**D**,**H**,**L**) are in Shandong Province.

Each cohort was numbered based on the number of fruiting branches where its first fruiting site was sorted into the cohort. Similarly to the boll distribution, the accumulation of

bolls by cohort was comparable between CT and MT under almost all situations (Figure 5), suggesting similar maturity in terms of boll formation. However, there was an obvious difference in 2020 between CT and MT in Hejian, where CT accumulated bolls more slowly than MT from the 3rd to the 10th cohort (Figure 5I).

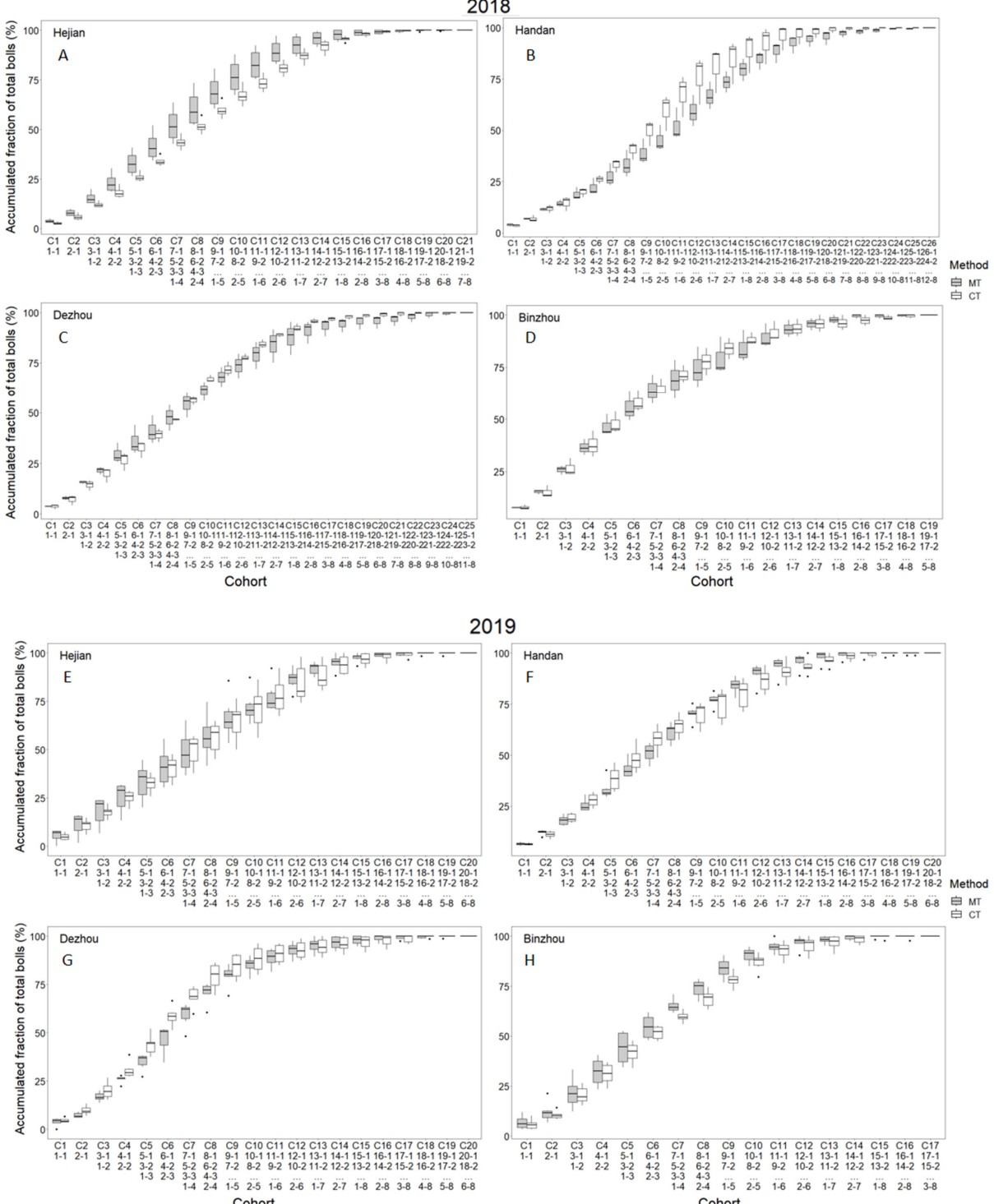

**Figure 5.** *Cont.*

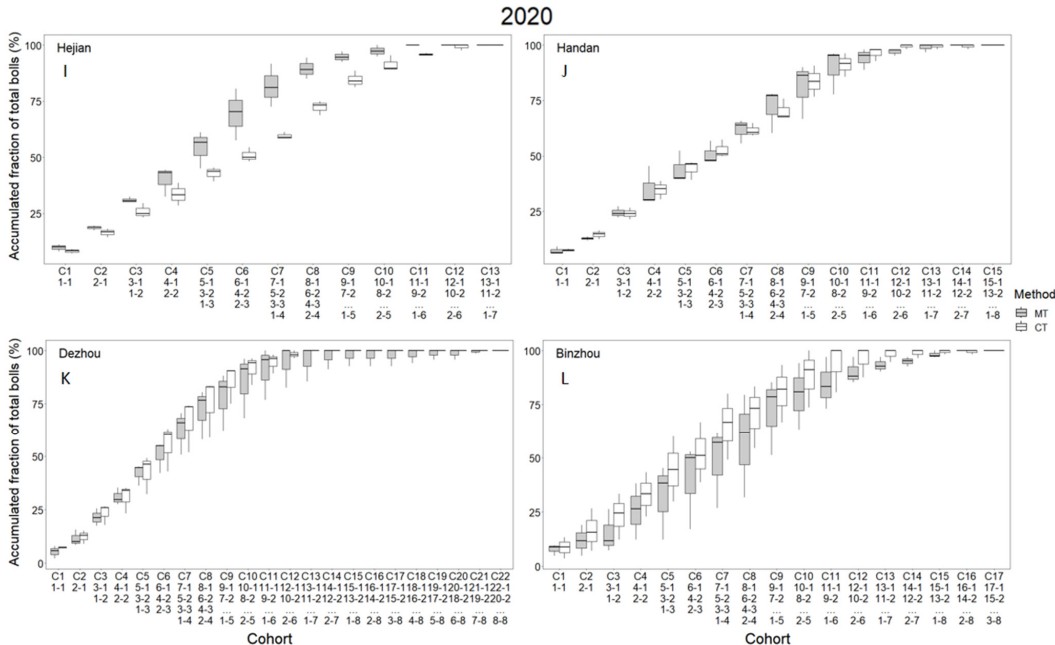

**Figure 5.** Effect of topping treatments during flowering on accumulated boll fraction by cohort (**C**) at Hejian (**A,E,I**) and Handan (**B,F,J**), Hebei Province, and Dezhou (**C,G,K**) and Binzhou (**D,H,L**), Shandong Province in 2018 (**A–D**), 2019 (**E–H**) and 2020 (**I–L**). Each cohort includes fruiting sites blooming simultaneously. 1-1 means the first fruiting position on the first fruiting branch, and so on. MT and CT indicate manual topping and chemical topping, respectively.

### 3.4. Boll Opening Percentage at Defoliation

There were great variations in BOP (from around 30% to 95%) prior to defoliation (in mid- or late September) among years and locations (Figure 5), which could be attributed to climate, variety, and management practices. According to ANOVA, these were significant between locations. BOP in Dezhou was always higher than in other locations. Despite the wide range of BOP, there were no significant differences between CT and MT, suggesting CT did not affect end-of-season maturity.

### 3.5. Yield and Yield Components

In 2018, CT did not influence seed cotton yield, except in Handan, where it caused significantly less boll and yield reductions (Table 2). In addition, CT better increased the boll density than MT in Dezhou in 2018, but there was no significant difference in actual seed cotton yield (Table 2). Since the same varieties were used in 2019 (Guoxin18-4) and 2020 (Guoxin 26), ANOVA was performed for the two years. There were significant differences in yield and its components among locations. CT, however, did not influence the yield-related parameters (Table 3). In addition, there were no significant interactions between location and topping treatment.

**Table 2.** Effect of manual (MT) and chemical topping (CT) on seed cotton yield and yield components in 2018 *.

| Location | Treatment | Boll Density (10⁴ Bolls ha⁻¹) | Average Boll Weight (g boll⁻¹) | Seed Cotton Yield (kg ha⁻¹) | Lint Percentage (%) |
|---|---|---|---|---|---|
| Hejian, Hebei | MT | 76.2 | 5.6 | 3098.8 | 44.2 |
| | CT | 98.3 | 6.0 | 3366.3 | 41.7 |
| Handan, Hebei | MT | 124.5 [a] | 5.9 | 5534.5 [a] | 43.3 |
| | CT | 86.3 [b] | 6.1 | 4290.7 [b] | 42.0 |
| Dezhou, Shandong | MT | 86.2 [b] | 5.5 | 3995.3 | 42.5 |
| | CT | 97.6 [a] | 5.8 | 3773.0 | 42.1 |
| Binzhou, Shandong | MT | 98.9 | 4.8 | 3708.8 | 44.0 |
| | CT | 97.4 | 5.1 | 3669.9 | 43.9 |

* Different letters indicate significant differences within the location according to *T* test at the 0.05 probability level.

**Table 3.** Effect of manual (MT) and chemical topping (CT) on seed cotton yield, and yield components in 2019 and 2020 *.

| Location | 2019 | | | | 2020 | | | |
|---|---|---|---|---|---|---|---|---|
| | Boll Density (10⁴ bolls ha⁻¹) | Average Boll Weigh t (g boll⁻¹) | Seed Cotton Yield (kg ha⁻¹) | Lint Percentage (%) | Boll Density (10⁴ bolls ha⁻¹) | Average Boll Weight (g boll⁻¹) | Seed Cotton Yield (kg ha⁻¹) | Lint Percentage (%) |
| Hejian, Hebei | 103.25 [a] | 5.68 [c] | 4510.5 [c] | 44.0 [ab] | 88.4 [b] | 5.66 [c] | 4424.0 [a] | 44.6 [b] |
| Handan, Hebei | 109.8 [a] | 6.07 [b] | 5741.2 [a] | 43.4 [b] | 112.65 [a] | 5.73 [c] | 4322.7 [a] | 42.0 [c] |
| Dezhou, Shandong | 67.25 [b] | 6.92 [a] | 4944.9 [b] | 45.1 [a] | 105.85 [a] | 6.58 [b] | 4448.8 [a] | 46.1 [a] |
| Binzhou, Shandong | 64.4 [b] | 6.20 [b] | 3716.7 [d] | 43.1 [b] | 75.4 [c] | 6.83 [a] | 3804.9 [b] | 46.5 [a] |
| **Treatment** | | | | | | | | |
| Manual topping | 89.3 | 6.12 | 4697.0 | 44.3 | 92.5 | 6.15 | 4238.7 | 44.9 |
| Chemical topping | 85.7 | 6.23 | 4726.1 | 43.5 | 97.6 | 6.17 | 4288.2 | 44.7 |
| ANOVA | | | | | | | | |
| Treatment | ns | ns | ns | ns | ns | ns | ns | ns |
| Location | <0.01 | <0.01 | <0.01 | 0.011 | <0.01 | <0.01 | 0.037 | <0.01 |
| Treatment × Location | ns | ns | ns | ns | ns | ns | ns | ns |

* Different letters indicate significant differences according to least significant difference at the 0.05 probability level.

## 4. Discussion

### 4.1. Boll Distribution, Accumulation, and Maturity Varied Greatly with Environment and Management Practices

As expected, boll distribution and cotton maturity differed extensively among years and locations. Compared with 2018 and 2019, the spatial scale of boll distribution was narrower in 2020 (Figure 4). The reasons may lie in the much lower precipitation in July and excessive precipitation in August 2020 (Figure 2), which caused bolls to concentrate on lower and middle fruiting branches, concurrent with a strong fruit abscission on upper fruiting branches. Consequently, the number of cohorts was lowest in 2020, except in Dezhou (Figure 5I–L).

Averaged across years, Dezhou had a higher cohort number (i.e., wider boll distribution) than other locations, which could be attributed to its higher soil fertility (the previous crop being a vegetable patch) and sparser population (Table 1) that resulted in more robust plants. For Binzhou, which is near the Bohai Sea (with only about 90 km distance from the sea), the number of cohorts was the lowest in 2018 and 2019, possibly due to the inhibition of plant growth by soil salinity.

The BOP in the late season in Hejian varied greatly among the years. It reached 85% on 22 September 2020, but only 28% on 20 September 2018 and 50% on 20 September 2019. This was associated with the severe fruit shedding on upper sympodial branches in 2020 compared with 2018 and 2019. In addition, Dezhou showed the highest BOP (96%) in late September 2019, also accompanied by complete fruit shedding on upper parts of plants, caused by the outbreak of bollworms in August and failure in chemical control.

### 4.2. Chemical Topping Did Not Affect Cotton Maturity Compared with Manual Topping

Regular multiple MC application (Table 1) was performed to control excessive growth in both chemical topping (CT) and manual topping (MT) plots, and the frequencies and rates were determined based on precipitation and plant growth. That is to say, CT uses an extra MC application (in addition to regular multiple MC application) to replace manual removal of plant apex. Some studies have shown that CT created a more compact plant type than MT [8,9], and thus improved the ventilation and light distribution within the canopy, which could enhance the opening of bolls and result in earliness [14,25,26]. In our previous study, we found that the late CT (seven days later than MT, near the physiological cutout, when the nodes above white flower are equal to 5.0) may delay maturity (characterized by BOP in late season) compared with early (at peak bloom) or middle CT (seven days later than peak bloom, at the same time as MT), and a high MC (98% soluble powder) rate (270 g ha$^{-1}$) used to execute CT may delay maturity than low (90 g ha$^{-1}$) or medium (180 g ha$^{-1}$) rates [19]. In this study, we further estimated how middle CT with medium MC rate affect cotton maturity that is measured during the season and at the end of the season based on data collected from four locations during 2018–2020. Firstly, we determined if CT alters the boll maturation period, but found no significant differences between CT and MT across the six environments (four in 2019 and two in 2020). Secondly, the spatial distribution and temporal accumulation of bolls were examined. Among 12 environments (three years × four sites), the difference in spatial boll distribution was observed only under three environments (Hejian in 2018 and 2019 and Handan in 2019), where CT produced taller but slender boll distribution by fruit positions than MT (Figure 4A,E,F). However, there were few differences in accumulated boll fraction by cohort (i.e., temporal distribution) between CT and MT under these three environments (Figure 5A,E,F). That was because the inner bolls on the upper part and outer bolls on the lower part may appear simultaneously. As a result, CT showed comparable BOP with MT in the late season, irrespective of year and location (Figure 6). Briefly, we provided comprehensive and definite evidence that CT performed at the same time as MT with medium MC rate (180 g ha$^{-1}$) did not influence cotton maturity, which negates the assumption of late maturity. Dai et al. (2022) also found that CT did not affect cotton maturity compared with MT [18].

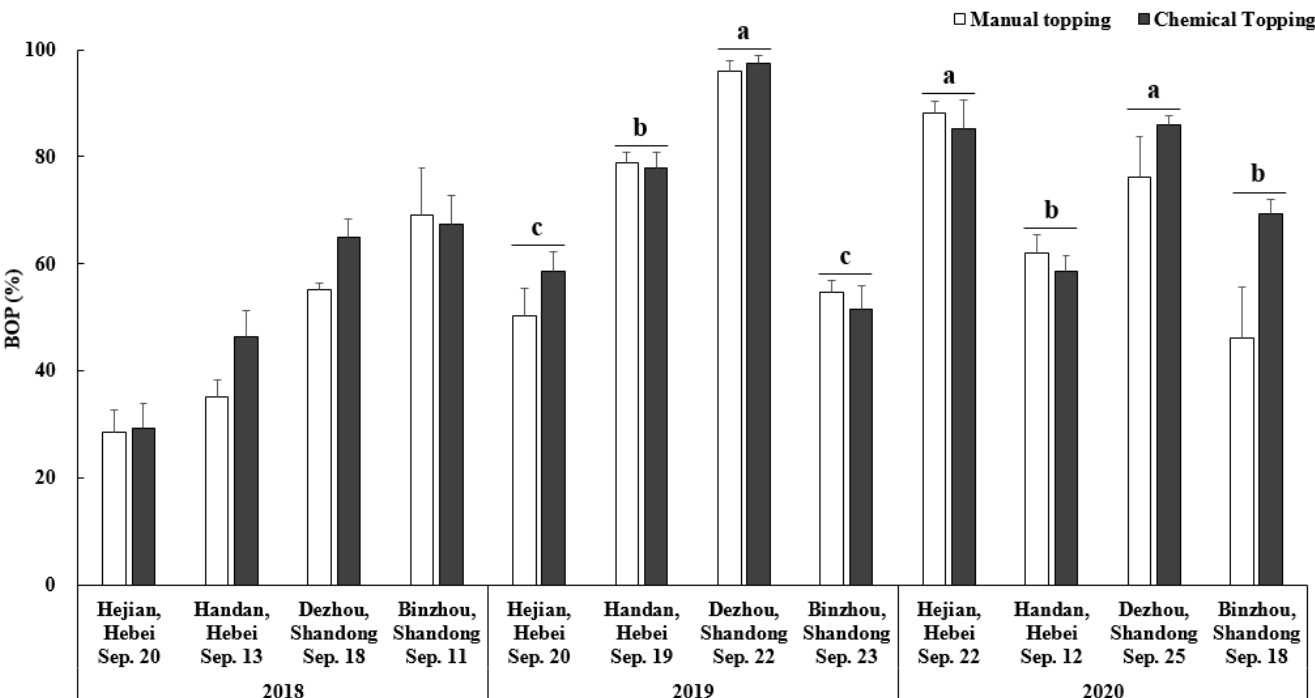

**Figure 6.** Effect of topping treatments during flowering period on boll opening percentage (BOP) in mid- or late September; *t*-tests were used to compare the means of manual topping and chemical topping at the 0.05 probability level. Different letters indicate significant differences among locations within the year according to least significant difference at the 0.05 probability level.

In addition, there was a clear difference in boll accumulation by cohort between CT and MT at Hejian in 2020 (Figure 5I), where CT showed lower accumulated boll fraction than MT from the 3rd to 11th cohort, which was mainly associated with the abscission at the first fruit site on the 5th–7th sympodium of CT-treated plants (Figure 4I). However, CT had the same accumulated boll fraction as MT at the 12th and 13th cohorts. This was because CT produced more bolls than MT at the first fruit sites of the 8th–10th fruiting branches, at the second fruit site of the 6th–8th fruiting branches, and the third fruit site of the 4th–5th fruiting branches (Figure 4I). Therefore, CT did not significantly influence the BOP on 22 September 2020 (Figure 6).

### 4.3. Chemical Topping Did Not Alter Yield Unless under Drought Stress

The data for 2018 were analyzed separately at each location due to the different varieties used. CT did not affect seed cotton yield at three of the four locations, and only reduced yield in Handan. From the climatic data, the precipitation in August 2018 in Handan was only 70.0 mm, much less than the local average for this month (119.8 mm). It is well established that cotton yield responds negatively to MC application under drought condition [27,28]. Therefore, it appears that CT with MC caused an over control of cotton plants in Handan in 2018, thereby reducing boll retention and yield.

In 2019 and 2020, there were significant variations in yield among locations, which was ascribed to the differences in environments and management practices. However, CT did not affect yield or yield components, and no interactions between topping treatment and location were found (Table 3). The yield results for the three years at the four sites (except in Handan in 2018) were in agreement with most reports on CT treatment in cotton [4,17,18].

### 5. Conclusions

In this study, we examined the effects of chemical topping (CT) with mepiquat chloride (MC) on cotton maturity based on boll age, spatial boll distribution, accumulation of boll fraction by cohort, and boll opening percentage in the late season, with manual topping

(MT) as control. The yield and yield components were also compared between CT and MT. The results suggest that CT did not influence cotton maturity or yield (except under drought), even if there were considerable differences among years and locations. Therefore, CT with MC (around 180 g/ha) performed during flowering like MT would be safe for cotton production in terms of maturity and yield. The results will be applicable to the areas with similar ecological conditions and social problems that urgently need to replace extensive manual labors by chemical practices, such as in the Yellow River Valley of China.

**Author Contributions:** Visualization, H.Q. and C.X.; conceptualization: X.T. and Z.L. (Zhaohu Li); data curation, H.Q., C.X., M.D. and X.T.; formal analysis, C.X. and X.T.; investigation, H.Q., W.Z., D.X., Z.L. (Zhengying Lu), R.S. and G.W.; project administration, Z.L. (Zhaohu Li); resources, W.Z., D.X., Z.L. (Zhengying Lu), R.S., G.W. and M.D.; supervision, M.D., X.T. and Z.L. (Zhaohu Li); writing—original draft, H.Q. and C.X.; writing—review and editing, A.E.E. and X.T. All authors have read and agreed to the published version of the manuscript.

**Funding:** This work was supported by the China Agriculture Research System (CARS-15-16).

**Data Availability Statement:** Data are contained within the article.

**Acknowledgments:** We thank Du Meng, Liao Bao-peng, Wei Ze-xin, Tan Zhi-xin, Wang Yu-kun, Xie Liu-wei and An Yang-yang of China Agricultural University for their assistance in data analysis.

**Conflicts of Interest:** The authors declare that they have no conflicts of interest.

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
