# Peer review of "Chemical Topping with Mepiquat Chloride at Flowering Does Not Compromise the Maturity or Yield of Cotton"

_agronomy, doi:10.3390/agronomy13020497_

Round 1

Reviewer 1 Report

The paper highlights an important assessment of alternative agronomist practices used in the particular region of China.  While the study has less relevance to other regions around the world it nether the less is shows how similar outcomes can be achieved with chemical applications versus the need for extensive manual labour.

The paper could benefit from the following:

1. The rates appear very high if the rates described in the methods are the active ingredient per ha.  Some mention of the actual product used or clearly specifying that the rates are in fact active ingredients levels would help.

2. Define boll fraction in the abstract.

3. In the introduction some mention of season length for the region compared to other regions around the world may help.  Manual topping in long season areas can actually lead to regrowth.  

4. Some discussion around the role of the other applications of MC (not at 5 nodes above white flower) would help.  If the same rates were applied to the Manual topping these may have have prevented differences in responses.  Some rationale of why various rates are chosen may also help explain some differences.

Reviewer 2 Report

This is a well written paper and interesting study, however I ask the authors to address two major concerns with the paper:

One relates to the hypothesis that mepiquat chloride will delay maturity, the literature and indeed cultural practice is to apply mepiquat chloride to prevent delays in maturity so I have difficulty in understanding that conclusion, there are significant studies investigating that response. I encourage the authors to do a wider review of the literature relating to this response and use of mepiquat chloride and address the difference in their hypothesis about delaying maturity to that in the literature.

With regards to the comparison of manual topping to the use of mepiquat chloride the authors have recently published a paper investigating that same question, it is unclear to me the difference between the two studies in terms of significant difference in novelty and innovation and find it unusual that the authors don't refer to this very recently published and extremely similar study. 

The authors will need to clearly articulate the difference between their two studies and the innovation or novelty of this work in the manuscript.

I have made one comment in the document relating to their inconsistent intepretation of results, in one section they attribute differences to mepiquat chloride reducing boll retention but in the paragraph above they have not drawn that same conclusion. 
